# Predicting partially observable dynamical systems via diffusion models with a multiscale inference scheme

**Rudy Morel**[*,1]**, Francesco Pio Ramunno**[2,3]**, Jeff Shen**[4]**,**
**Alberto Bietti**[1]**, Kyunghyun Cho**[5]**, Miles Cranmer**[6]**, Siavash Golkar**[1,5]**, Olexandr**
**Gugnin**[7]**, Geraud Krawezik**[1]**, Tanya Marwah**[1]**, Michael McCabe**[1,5]**, Lucas Meyer**[1]**,**
**Payel Mukhopadhyay**[6,8]**, Ruben Ohana**[1]**, Liam Parker**[1,8]**, Helen Qu**[1]**, François Rozet**[9]**,**
**K.D. Leka**[10,11]**, François Lanusse**[1,12]**, David Fouhey**[5]**, Shirley Ho**[1,4,5]

**The Polymathic AI Collaboration**

[1]Flatiron Institute, [2]University of Geneva, [3]FHNW, [4]Princeton University, [5]New York
University, [6]University of Cambridge, [7]University of Kyiv, [8]University of California,
Berkeley, [9]University of Liège, [10]NorthWest Research Associates, [11]Nagoya
University, [12]Université Paris-Saclay, Université Paris Cité, CEA, CNRS, AIM.

## Abstract

Conditional diffusion models provide a natural framework for probabilistic prediction of dynamical systems and have been successfully applied to fluid dynamics and weather prediction. However, in many settings, the available information at a given time represents only a small fraction of what is needed to predict future states, either due to measurement uncertainty or because only a small fraction of the state can be observed. This is true for example in solar physics, where we can observe the Sun's surface and atmosphere, but its evolution is driven by internal processes for which we lack direct measurements. In this paper, we tackle the probabilistic prediction of partially observable, long-memory dynamical systems, with applications to solar dynamics and the evolution of active regions. We show that standard inference schemes, such as autoregressive rollouts, fail to capture long-range dependencies in the data, largely because they do not integrate past information effectively. To overcome this, we propose a multiscale inference scheme for diffusion models, tailored to physical processes. Our method generates trajectories that are temporally fine-grained near the present and coarser as we move farther away, which enables capturing long-range temporal dependencies without increasing computational cost. When integrated into a diffusion model, we show that our inference scheme significantly reduces the bias of the predicted distributions and improves rollout stability.

## 1 Introduction

Probabilistic prediction of dynamical systems is at the heart of many challenging tasks in science and engineering. Diffusion models have recently shown success in probabilistic prediction for physical systems, especially when they are applied to simulated environments [39] or to settings such as terrestrial weather prediction [62], where laboratory settings or advanced data assimilation can recover much of the current system state [27].

Many real systems are *partially observable*, meaning that data is missing, unobtainable, or sufficiently noisy such that at any given time there is inadequate information to accurately infer the underlying state of the system. It follows, then, that there is inadequate information to predict its exact evolution. In these settings, the correct incorporation of past information can help predict future trajectories.

---

[*]Contact: rmorel@flatironinstitute.org

39th Conference on Neural Information Processing Systems (NeurIPS 2025).

A prime example of such a partially observable system is our nearest star. Key components governing the dynamics of the Sun are not directly observable (e.g, the driving forces beneath the visible "surface"), and what *is* observable is only available via remote sensing. Nonetheless, predicting this particular system's evolution is important due to the potential impact on technology-based sectors of society arising from solar energetic events [56]. While domain experts have identified physical descriptors associated with energetic phenomena such as solar flares [40, 10, 42], and relevant ML-ready datasets have been curated and published [22, 4, 16], there does not yet exist a model (physics-based or ML-based) that can predict future states of solar active regions and their magnetic fields across the spatial and temporal scales relevant to significantly improve prediction for these events [41, 5].

In this paper we study the problem of predicting partially observable dynamical systems with diffusion models [29], motivated by the challenging problem of learning solar dynamics from data. As a benchmark to encourage community progress on this problem, we assemble an 8.5TB dataset of $512 \times 512$ videos of solar regions containing 12 fields with measurements of the magnetic vector field and the Sun's atmosphere. Diffusion models developed for well-observed fluid simulations [39] or reanalyzed terrestrial weather data [62] typically use an autoregressive inference scheme to generate future predictions, conditioning on only a few past frames (typically two). For solar dynamics, however, we find that such models struggle to accurately predict the evolution, showing significant deviation from observations over time.

To address these limitations, we introduce a new multiscale inference scheme based on "multiscale templates", which provide an efficient way to condition on distant past information without increasing computational cost. These templates enable the generation of distant future time steps while conditioning on fine-grained present information and coarse-grained past times. A model trained on generating such videos can then be used to generate arbitrarily long trajectories in the future, by combining different multiscale templates. Compared to inference schemes such as standard autoregressive rollouts used in the literature [39, 62], our method predicts a distant future time step from past observations in a single call to the diffusion model, avoiding the accumulation of distribution errors. Furthermore, we condition more frequently, and on a larger portion of past observed data.

**Contributions.** Our key contributions are: **(a)** We introduce a new multiscale inference scheme tailored to partially observable dynamical systems encountered in Physics. **(b)** On the challenging task of solar prediction, our multiscale inference scheme outperforms standard schemes from the literature on diffusion models for physics and natural videos, reducing prediction bias and instability. **(c)** To the best of our knowledge, our model is the first multi-modal diffusion model trained to predict high-resolution solar videos; prior work focuses on single modality, low-resolution data (both in time and space). **(d)** To encourage competition on the challenging problem of solar prediction, we provide a new multi-modal 8.5TB dataset of $512 \times 512$ videos capturing solar regions. Upon publication, our dataset and model will be made publicly available.

## 2   Related works

**Diffusion models for predicting dynamical systems.** Unlike [47, 60], which employ a diffusion model to learn the distribution of individual states in order to refine predictions from a predictor network, our work falls within the scope of modeling the dynamic of the observations. Along these lines, [39, 69] address highly observable dynamical systems, like fluids governed by the Navier–Stokes equations, where all relevant variables (e.g., velocity, pressure) are accessible. Other works [62] train on data from complex reanalysis of sparse observations (e.g., the ERA5 dataset [27]). Full observation or re-analysis is not always feasible. For instance, in solar dynamics, it is challenging to accurately recover surface observations at even moderate scales [see, e.g. 7, 14], and becomes especially difficult when attempting to infer the state of the Sun's interior [64, 49], energy transfer [82] or forces acting on the plasma [11, 88], yet this information is key to predicting solar dynamics. Thus, while [39, 62] see no benefit from using more than two past observations, incorporating additional past steps substantially improves results in our setting. In that sense, our findings align with those of [70] even though they focused on deterministic models. Diffusion models can perform data assimilation and prediction from incomplete observations simultaneously [66, 74, 33], but this requires a dataset of underlying system states to train the model – an assumption we do not make in this paper.

**Inference schemes for diffusion models.** The standard autoregressive inference scheme for video diffusion [31, 8, 26, 23, 68] consists in generating progressively an entire video by sliding a short window. Beyond this, Flexible Diffusion Models (FDM) [25, FDM] and Masked Conditional Video Diffusion [84] both adopt flexible conditioning strategies and train a single model with a randomized masking. In particular, [25] introduces two types of inference schemes. The first, called "long-range," generates progressively more distant future frames while conditioning only on recent ones, thereby discarding distant past information. The second, called "hierarchy-2," uses a sliding-window with an initial long-range prediction, but it conditions on past information only at the first iteration. In contrast, our multiscale inference scheme generates videos at multiple scales and conditions on past information across multiple iterations, which is crucial for recovering information in partially observable dynamical systems.

**Machine learning for solar physics.** Machine learning is increasingly used across heliophysics [13, 5], in particular for predicting solar energetic events [9, 57, 59, 58, 20, 44]. However, these approaches typically perform classification based on selected features rather than modeling the temporal evolution of the solar atmosphere. Other works apply ML to enhance data quality [12, 35, 86, 34, 24] or build large-scale pretrained models [85], but these also do not predict future physical states. When it comes to predicting future solar trajectories, many works either focus on a single quantity of interest [6, 65, 21] or operate on limited spatiotemporal resolutions. For example, [65, 1] use at least a $4\times$ spatial downsampling factor and a temporal resolution no finer than 12h. In contrast, our dataset uses multiple modalities (associated to different instruments); is downsampled only $2\times$ spatially, matching the optical resolution of the instrument; and is captured at 1h sampling rate.

## 3 Background: Conditional Diffusion models

This section presents the aspects of conditional diffusion models [75, 29] most relevant to our work.

**Score-based diffusion model.** Score-based generative models [77, 78], are a class of generative models that learn to sample from complex data distributions by reversing a gradual noising process. These models define a forward diffusion process in which the input data $\mathbf{x} \in \mathbb{R}^N$ is progressively corrupted by adding Gaussian noise at various noise levels $\sigma_s$

$$\mathbf{x}_s = \mathbf{x} + \sigma_s \boldsymbol{\epsilon} \ , \ \boldsymbol{\epsilon} \sim \mathcal{N}(0, I_N). \tag{1}$$

The resulting distribution over the noisy data is denoted by $p_s(\mathbf{x}_s)$ and captures how the original data distribution evolves under increasing noise. The generative model learns a reverse denoising process which maps a Gaussian distribution to the distribution of the data [77, 3, 81]. This can be described as a stochastic differential equation

$$d\mathbf{x}_s = -\sigma_s^2 \nabla \log p_s(\mathbf{x}_s) ds + \sigma_s dW_s, \tag{2}$$

and involves the score function $\nabla \log p_s(\mathbf{x}_s)$. This score can be obtained by solving a denoising task [29, 77, 78, 46, 76, 37]. Indeed, if we write $D(\mathbf{x}, s)$ a function that minimizes the $L^2$ loss

$$\mathbb{E}_{\mathbf{x} \sim p_{\text{data}}, \boldsymbol{\epsilon} \sim \mathcal{N}(0, I_N)} \left[ \|D(\mathbf{x}_s, s) - \mathbf{x}\|^2 \right] \ , \ \text{with } \mathbf{x}_s = \mathbf{x} + \sigma_s \boldsymbol{\epsilon}. \tag{3}$$

then we can show [83, 18, 38, 52] that the score is given by $\nabla \log p_s(\mathbf{x}_s) = (D(\mathbf{x}_s, s) - \mathbf{x}_s)/\sigma_s^2$.

Therefore, a diffusion model is trained by learning a neural network $D_\theta$ with parameters $\theta$ on the denoising loss (3), and sampled by discretizing the reverse process (2).

**Conditional diffusion model.** In the paper, beyond modeling the distribution $p(\mathbf{x})$ of the data, we focus on modeling conditional distributions $p(\mathbf{x}|\mathbf{y})$ where $\mathbf{x}$ is a trajectory and $\mathbf{y}$ is a part of the trajectory itself [84, 67]. To that end, let $\boldsymbol{m} \in \{0, 1\}^N$ denote a vector (or *mask*) indicating which parts of the signal $\mathbf{x}$ are used as conditioning. The conditioning data is written $\boldsymbol{m} \odot \mathbf{x}$, where $\odot$ is the element-wise product. As above, the distribution $p(\mathbf{x}|\boldsymbol{m} \odot \mathbf{x})$ can be modeled by learning a denoiser to reconstruct the "clean" data $\mathbf{x}$ from its noised version $\mathbf{x}_s$ with in addition the information of the conditioning:

$$\mathbb{E}_{\mathbf{x} \sim p_{\text{data}}, \boldsymbol{\epsilon} \sim \mathcal{N}(0, I_N)} \left[ \|D((1 - \boldsymbol{m}) \odot \mathbf{x}_s + \boldsymbol{m} \odot \mathbf{x}, s, \boldsymbol{m}) - \mathbf{x}\|^2 \right] \ , \ \text{with } \mathbf{x}_s = \mathbf{x} + \sigma_s \boldsymbol{\epsilon}. \tag{4}$$

where the mask $\boldsymbol{m}$ is fed to the denoiser $D_\theta$ to help differentiate between noised data and conditioning data. This way, the denoiser is trained to retrieve the global noise from the noised data $\mathbf{x}_s$ just like Eq.(3), but with additional conditioning clean information $\boldsymbol{m} \odot \mathbf{x}$.

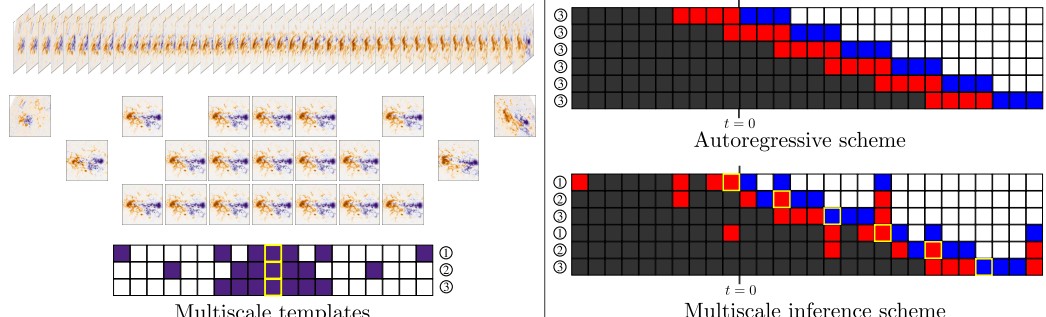

Figure 1: **Multiscale templates and inference scheme. (Left)**: Our multiscale templates in **purple**. **(Right)**: Comparing a standard autoregressive scheme (on top) with our multiscale inference scheme. We use the visualization style of [25], in which **dark** boxes indicate available steps (either observed or generated at previous iterations) and **red** and **blue** boxes indicate steps that are used as conditioning or generated, respectively. Each row is a new call to the conditional diffusion model with the used template indicated by the number next to the row. Our inference scheme enables capturing longer-range dependencies, conditions more often in the past, and mitigate rollout instability by generating a distant future (9 on the figure) in one call to the conditional diffusion model.

## 4 Multiscale inference scheme for physical processes

In this paper, we are interested in predicting a dynamical system from its observations $\mathbf{x}$, e.g. the magnetic field at the surface of the Sun. At each time $t$, we denote $\mathbf{x}_t$ the observation of the system, which provides only a partial view of the underlying true state.

At present time $t = 0$, the goal is to generate a future realization $\mathbf{x}_{1:T}$ at horizon $T$ conditionally on the past $\mathbf{x}_{t \leq 0}$. In doing so we aim to approximate the following conditional distribution

$$p(\mathbf{x}_{1:T}|\mathbf{x}_{t \leq 0}). \tag{5}$$

Due to computational constraints, modeling the full distribution over long horizons $T$ is infeasible. A common approach is to compress the data to extend the effective context length, as done in latent diffusion models [8, 26, 23], but the question remains, how to generate arbitrarily long trajectories using a generative model with a fixed trajectory length?

We assume that our conditional diffusion model can generate only a subset of $2K + 1$ time steps at once. We seek to use the fixed-size model to produce samples over a far larger set of $T \gg 2K + 1$ steps by repeatedly applying the fixed length model. For convenience, assume that the model always generates $K$ future steps from white noise, and the remaining $K + 1$ are conditioning (from the past or present). Generating a trajectory of length $T$ thus requires at least $\lceil T/K \rceil$ steps. If we define $I_n$ as the set of $K$ new time indices generated and $C_n$ the set of $K + 1$ frames used as conditioning, the iterated process amounts to the following approximation:

$$p(\mathbf{x}_{1:T}|\mathbf{x}_{t \leq 0}) \approx \prod_{n=1}^{N} p(\mathbf{x}_{I_n}|\mathbf{x}_{C_n}). \tag{6}$$

A collection of pairs of index sets $(I_n, C_n), 1 \leq n \leq N$, is called an *inference scheme*. Given the above fixed budget constrain, these sets must satisfy $|C_n| = K + 1, |I_n| = K$. We write $P_n$ the set of indices available at step $n$, which is defined recursively as $P_1 = \{t \leq 0\}$ (observed past) and $P_n = P_{n-1} \cup I_n$ (available time steps). To properly formalize the problem, we consider inference schemes that satisfy the following properties:

- **(completeness)** $\cup_{n=1}^{N} I_n = \{1, \ldots, T\}$

- **(admissibility)** $C_n \subset P_n$, the conditioning is done on already generated (or observed) steps

- **(efficiency)** $I_k \cap I_\ell = \emptyset$ for $k \neq \ell$, no future step is generated twice

For example, an autoregressive inference scheme consists of sliding a fixed-size fine-grained window progressively forward in time, $C_n = \{(n-1)K, \ldots, nK\}$ and $I_n = \{nK + 1, \ldots, (n+1)K\}$

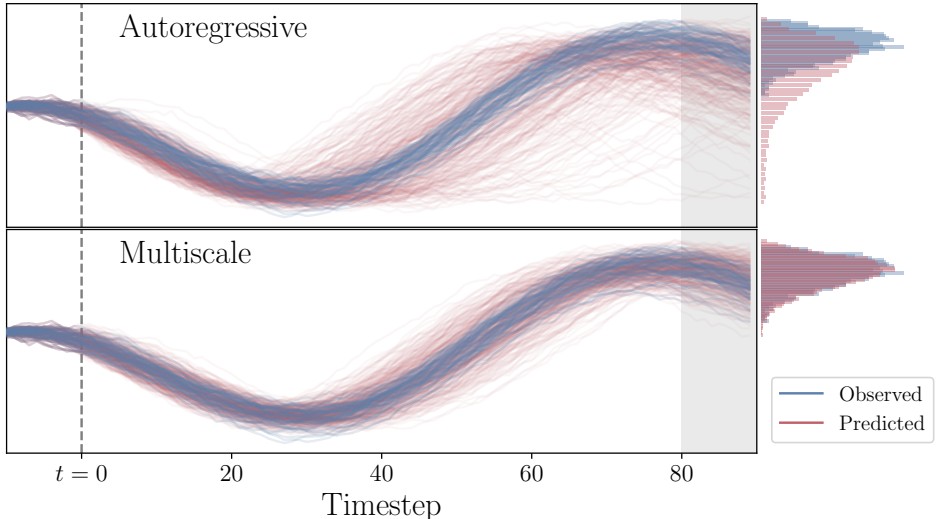

Figure 2: **Performance of our multiscale inference scheme on a synthetic example.** The observed data (blue) consists of Gaussian fluctuations around a sinusoidal trend. Predictions (red) are from a diffusion model with access only to past data $t \leq 0$. **(Top):** The global trend is barely observable at fine scale. Thus, a model that generates small trajectory segments autoregressively tends to accumulate errors, leading to biased and overly broad predicted distributions. **(Bottom):** Our multiscale inference scheme (see Fig. 1) efficiently recovers the target distribution – with a Wasserstein distance of 0.021 vs. 0.23 for the autoregressive model. When restricted to the same 3-step past horizon, the multiscale inference still performs better, with a Wasserstein distance of 0.08.

as shown on Fig. 1. This autoregressive inference scheme has several downsides, as evidenced in Tab. 1 and illustrated in Fig. 1. The main one being that after the second iteration, there is no explicit conditioning on observed data, which contributes to rollout instability.

## 4.1   Multiscale templates for physical processes

Finding an appropriate inference scheme for partially observable dynamical systems is challenging due to the large space of possibilities: many candidates exist for pairs of conditioned times $C_n$ and generated times $I_n$ at each step that satisfy the above properties.

To guide our design, we highlight two key challenges encountered in predicting physical systems:

(a) **Partially observable.** The state of the system at any given time cannot be fully determined from the observations. Consequently, the distribution of future scenarios conditioned on past observed data may not be restricted to a Dirac measure. In many cases, the system state cannot be fully observed due to missing measurements of key physical variables (e.g., velocity fields, or unresolved structures), insufficient observational resolution, or corruption arising from instrumental noise.

(b) **Long-memory.** Many physical processes exhibit long memory, or long-range dependency, in time. This can be quantified by a smooth decay of the autocorrelation (sometimes characterized in the frequency domain by a power-law decay of the power spectrum [51, 79, 2, 48, 53, 55]). Intuitively, observations closer to the present have a stronger impact on the future and the influence of distant past observations gradually diminishes while remaining significant.

Diffusion models have been applied to predicting dynamical systems without fully addressing challenge (a) or relying on additional information to overcome it. For example, [39] apply a diffusion model to fully resolved fluids which are effectively Markovian. In weather prediction, although the observed data is sparse, data assimilation—also known as reanalysis—enables the reconstruction of missing information, resulting in large datasets of highly informative states [27], on which diffusion

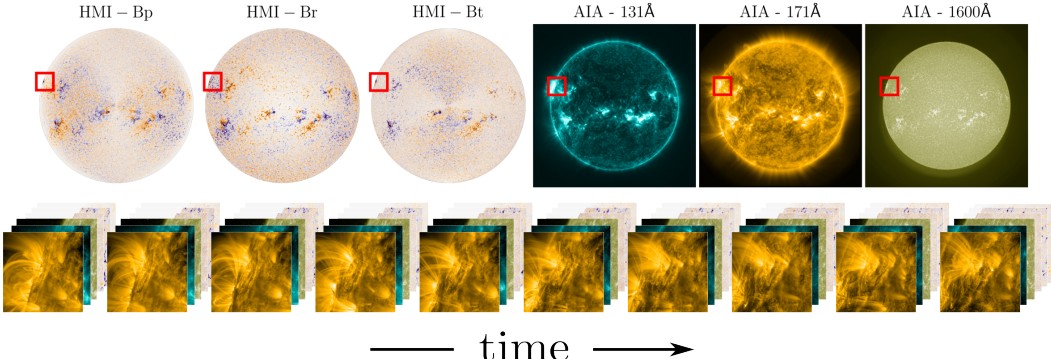

Figure 3: **(Above):** Example full-disk solar images from 2015-12-12 (see § 5.2 for details). The left three panels are photospheric vector magnetic-field components; the right three panels are images of the solar corona and chromosphere. "Active regions" (intense magnetic fields connected to bright coronal structures) are present in both modalities. **(Below):** A sequence of frames of a cropped active region, corresponding to the red box in the row above.

models have been successfully trained [62]. Other models handle missing states, but require clean sates for training [74, 33], which is not always available.

In this paper, we tackle the challenging problem of predicting the observations of a dynamical system presenting the two challenges (a) and (b) simultaneously, as is common across many disciplines. For example, in oceanography and climatology, shallow ocean layers are observed while few observations exist for the deep ocean [45]; and in seismology, subsurface stress is not directly measured [36]. In solar physics, the goal of predicting a future trajectories of active solar regions from available observations (of the magnetic solar surface and hot coronal atmosphere) is challenged by: (a) missing key components of the sate – in this case, observations of the interior of the Sun, with instrumental noise present in the data [32, 72], which is sometimes not fully understood or mitigated [73]. And (b), the targets that are of predictive interest, e.g., sunspots, have long-range dependencies described by plasma diffusion and flow patterns on local, moderate, and global spatial scales [14].

In principle, if the system state was knowable and described by well-constrained partial differential equations (*e.g.*, a magneto-hydrodynamic framework [63]), one could solve the dynamics forward in time from a single time step (Markov process). Now, under assumption (a), even if the underlying system is Markovian, its observations may not be predicted deterministically because of the lack of information; such systems are often called hidden Markov [19]). The combination of properties (a) and (b) as it is often the case in real cases, encourages a diffusion model to consider not only information near the present but further back in time to access what is needed to predict the future. Inspired by works on long-range temporal processes [2, 48, 55] and wavelets [51, 79, 15, 54], we introduce a framework to do this.

A *multiscale template* $\mathbb{T}_K^\alpha$ is a set of $2K + 1$ indices centered at the present $t_0^\alpha = 0$ and becoming progressively coarser farther from it, defined using time increments as powers of $\alpha \geq 1$:

$$\mathbb{T}_K^\alpha = \{t_{-K}^\alpha, \ldots, t_0^\alpha, \ldots, t_K^\alpha\} \text{ with } t_{k+1}^\alpha = t_k^\alpha + \alpha^k \text{ and } t_{-k}^\alpha = t_{-k+1}^\alpha - \alpha^k \tag{7}$$

This set of indices is symmetrical in $t_0^\alpha = 0$. For $\alpha = 1$, we retrieve a standard uniform window used in an autoregressive scheme. When $\alpha > 1$, the time indices are progressively more spaced as we move away from present. We allow $\alpha$ to be real, in that case, the template is mapped to integers through $\mathbb{T}_K^\alpha = \{\text{sign}(t_k^\alpha)\lfloor|t_k^\alpha|\rfloor, \ -K \leq k \leq K\}$ where $\lfloor t \rfloor$ is the integer part of $t$.

For a fixed budget of $K$ times, a multiscale template allows to consider a horizon in the past (and in the future), that is exponential in $K$, while a uniform template $\alpha = 1$ has a horizon that is linear in $K$. As we will see in the next section, this is crucial for capturing long-range dependencies, and helps stabilize long predictions.

The term *template* reflects the flexibility to later separate it into conditioning $C_n$ and newly generated time indices $I_n$ as needed, that is, to apply an arbitrary conditioning mask $\boldsymbol{m}$ in Eq. (4).

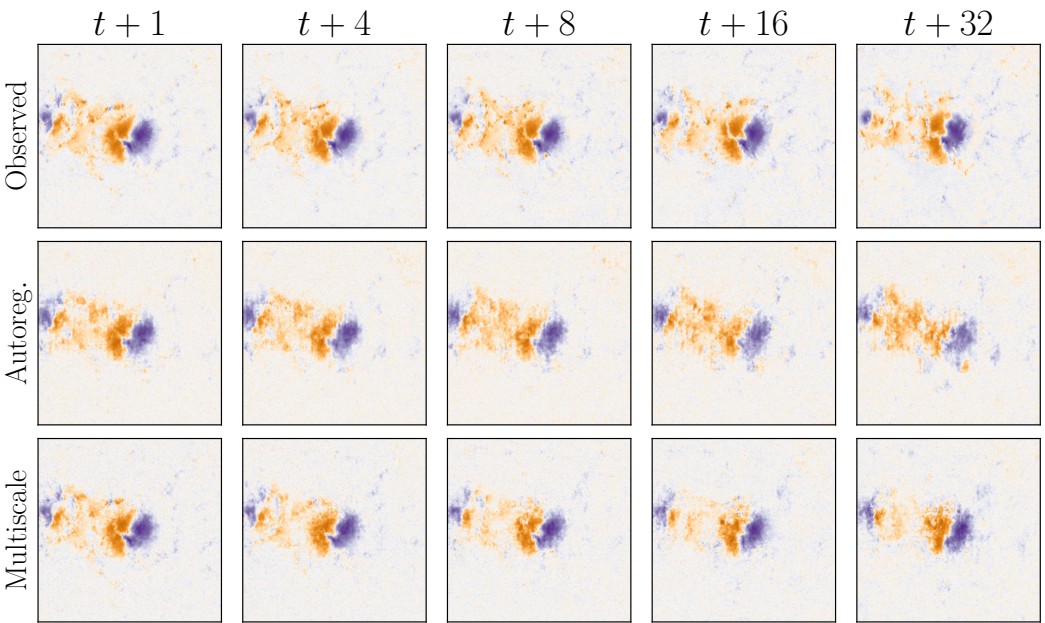

Figure 4: **Example of predictions, for different inference schemes:** autoregressive and multiscale (ours). Colorbar: -3000 ████ ████ 3000 Gauss (magnetic field).

## 4.2 Multiscale inference scheme

We now design an inference scheme to produce arbitrarily long future trajectories, using the multiscale templates introduced above and motivated by the key properties of observed physical systems. As described above, this involves defining pairs $(C_n, I_n)$ of conditioning indices and newly generated indices at each iteration $n$, that is, at each call to the diffusion model, which progressively cover a future trajectory (see Eq. (6)). In the experiments we choose to generate $K = 3$ new time steps at each iteration, which means our diffusion models generate small videos of length $2K + 1 = 7$, and we choose to use templates $\mathbb{T}_K^{\alpha_{\max}}$ with a maximum $\alpha_{\max} = 2.5$ (see Fig. 1); in the following we drop the dependence on $K$ and write $\mathbb{T}^\alpha$ directly. This means that the most extended video we will generate at once goes up to $9 = \lfloor 1 + 2.5 + 2.5^2 \rfloor$ steps in the past and future (see Eq. (7)). We refer the reader to the Appendix for multiscale inference schemes with different choices of $K$ and $\alpha_{\max}$.

Our inference scheme, illustrated in Fig. 1, begins by using the largest template $\mathbb{T}^{\alpha_{\max}} = \{-9, -3, 1, 0, 1, 3, 9\}$ to generate $K = 3$ steps in the future: $I_1 = \{1, 3, 9\}$ and conditioning on the $K + 1 = 4$ observed steps $C_1 = \{-9, -3, -1, 0\}$. This enables the model to generate the 9[th] step into the future while incorporating observed data that extends equally far into the past. Without completing an entire trajectory, this first step gives us predictions of the physical system at multiple horizons in the future. Once this multiple-horizon prediction is performed, the goal is to "fill the gaps" in the future using the other, shorter-range templates.

Then, we iterate over all possible templates $\mathbb{T}^\alpha$ with $1 \leq \alpha \leq \alpha_{\max}$ in decreasing order, along with all their possible shifts into the future. For each candidate, we check whether the shifted template overlaps with at least $K + 1 = 4$ available time steps. This ensures sufficient conditioning data to generate $K$ new steps. Among the valid options, we select the first template and shift whose final index aligns with the current maximum horizon, which is 9 in our experiments. This ensures that the generation proceeds in a consistent way, gradually filling in missing future steps while maintaining coherence with earlier generated data. In the experiments, we get $\mathbb{T} = \{-6, -2, -1, 0, 1, 2, 6\}$ which must be shifted by 3 steps in the future. The overlap with the previously generated time steps defines $C_2 = \{-3, 1, 3, 9\}$ and the newly generated indices at this second iteration are $I_2 = \{2, 4, 5\}$.

We repeat this procedure until all the gaps from the first applied largest template are filled. For the values chosen in the experiments, this requires applying a last multiscale template $\mathbb{T}^\alpha =$

Table 1: **Predictions performance.** We compare different inference schemes (Autoregressive, Hierarchy-2 [25], Ours – Multiscale) and models (AViT [50],AR-diff [39], Ours). For each, we evaluate at three different time windows (1-4 hours, 4-16 hours, 16-32 hours) using multiple metrics: the Wasserstein distance between the distributions; mean absolute error in the power spectrum; and normalized mean absolute error of representative solar physics quantities from [10] – the Mean Horizontal Gradient of the Total Field (MeanGBT) and of the Vertical Field (MeanGBZ).

| Model | Scheme | Wasserstein | | | MAE Power Spec. | | | NMAE MeanGBT | | | NMAE MeanGBZ | | |
|---|---|---|---|---|---|---|---|---|---|---|---|---|---|
| | | 1:4 | 4:16 | 16:32 | 1:4 | 4:16 | 16:32 | 1:4 | 4:16 | 16:32 | 1:4 | 4:16 | 16:32 |
| DiT | Autoreg. | 3.9 | 5.6 | 7.9 | 0.25 | 0.36 | 0.53 | 0.18 | 0.30 | 0.37 | 0.15 | 0.25 | 0.31 |
| DiT | Hiera. [25] | **3.0** | 4.6 | 6.0 | **0.12** | 0.27 | 0.38 | **0.12** | 0.28 | 0.38 | **0.09** | 0.22 | 0.31 |
| DiT | Ours | **3.0** | **4.3** | **5.5** | **0.12** | **0.22** | **0.33** | 0.14 | **0.27** | **0.33** | 0.10 | **0.21** | **0.27** |
| [50] | Autoreg. | 12 | 13 | 15 | **0.11** | 0.35 | 0.81 | 0.40 | 0.44 | 0.45 | 0.40 | 0.43 | 0.44 |
| [39] | Autoreg. | 7.3 | 12 | 16 | 0.20 | 0.47 | 0.71 | 0.29 | 0.52 | 0.67 | 0.27 | 0.49 | 0.64 |
| DiT | Ours | **3.0** | **4.3** | **5.5** | **0.12** | **0.22** | **0.33** | **0.14** | **0.27** | **0.33** | **0.10** | **0.21** | **0.27** |

$\{-3, -2, -1, 0, 1, 2, 3\}$, which is actually a uniform template, shifted by 6 in the future, and conditioned on the time steps $C_3 = \{3, 4, 5, 9\}$ and generating new time steps $I_3 = \{6, 7, 8\}$.

Once the first template span has been entirely generated, we shift the current present to the last generated step, 9 in the experiments, and can now repeat the above scheme to predict a complete video until 18 and so on (see Fig. 1).

This inference scheme offers key advantages. Compared to standard autoregressive or "hierarchy-2" schemes [25], it conditions more often on distant past and future information, better capturing long-range dependencies around the present. It predicts up to 9 steps ahead in a single diffusion call, whereas autoregressive methods require 3 calls for the same horizon. This improves error accumulation, though errors can still grow beyond the largest template's time scale.

The horizon of the largest template is chosen to be 9 in experiments but it can be adjusted (see Appendix for a general algorithm). If the physical process exhibits a finite decorrelation timescale, it is natural to choose a largest template that spans this timescale to fully capture long-range dependencies and mitigate rollout instabilities. We refer the reader to the Appendix for multiscale inference schemes based on larger templates.

## 5 Numerical experiments

### 5.1 Synthetic example

We present a synthetic example of time-series of observations $x_t = \mu_t + \eta_t$, where $\mu_t$ is a deterministic sinusoidal trend, and is made partially observable by the addition of Gaussian noise $\eta_t$. In the absence of noise, a single time step suffices to determine the future trajectory completely. In the presence of noise, however, consider the times around a negative peak (approximately $t = 30$; see Fig. 2). Depending on the noise realization, the local trend may be upward or downward, making the state difficult to recover locally. That is, partial observability induced by noise prevents accurate estimation of the underlying slowly varying component. It is thus necessary to look further into the past, which is precisely what our multiscale inference scheme achieves.

Fig. 2 shows predictions with a small diffusion model, with either an autoregressive scheme or our multiscale inference scheme. Our scheme better captures the trajectories than the autoregressive one, as confirmed visually and by Wasserstein distance (0.021 vs 0.23). Because of the partial observability of the trend mentioned above, the autoregressive scheme produces errors that accumulate.

Our multiscale scheme efficiently captures long-range dependencies through its multiscale templates (see Section 4.1). When predicting the future at $t = 0$, it also conditions on earlier steps (up to $-9$) compared to only $-3$ for an autoregressive scheme (see Fig.1). To isolate the effect of the multiscale template from that of conditioning further in the past, we restrict our scheme in Fig.2 to the same past

horizon ($-3$). Performance slightly degrades (from 0.021 to 0.08), but still surpasses autoregressive baselines.

We refer the reader to the Appendix for another synthetic example of a partially observable fluid dynamical system.

## 5.2 Solar dynamics prediction

**Solar dataset.** To encourage competition on predicting partially observable long-memory dynamical systems, we introduce a new ML-ready dataset (see Fig. 3) of reasonably high-resolution solar dynamics prediction based on real observations from the NASA Solar Dynamics Observatory mission [61], in continuous operation since 2010. The data contains two modalities from two instruments, surface magnetic fields [71], and images of the solar atmosphere [43]. Each produces $4096 \times 4096$-pixel images of the full disk of the Sun (see Fig. 3) at high cadence, making the data-handling very demanding. As discussed in [16], because active regions occupy only a small fraction of the visible disk, we propose a dataset of square-image videos of $512 \times 512$-pixel windows that track an active-region. This data is curated to carefully account for the rotation of the Sun, the limb of the Sun (its "edge"), co-alignment between the two modalities, potential overlap between targets, and uncertainty, artifacts, and missing data. Each day, we randomly sample 8 regions of the Sun to follow for 48h, sampled hourly. The regions are selected to avoid bias towards rare events. Our dataset consists of 8.5TB composed of $\approx$ 15K multi-channel videos of shape $48 \times 12 \times 512 \times 512$. Each video contains 3 magnetic fields channels and 9 channels for the solar atmosphere at different wavelengths. In the following, all models are trained on images downsampled by a factor of 2 (to the instrument's optical resolution) and considering only 3 of the atmosphere channels, in order to reduce the computational cost of training multiple diffusion models.

**Diffusion model hyperparameters.** We adopt a Vision Transformer [17, ViT] architecture as our denoiser backbone, following the approach in [37], but extended to handle spatio-temporal data and inspired by the implementation in [67]. The denoiser takes as input 3D patches of size $1 \times 8 \times 8$ (no patchification in time), and consists of 16 attention-based layers with a hidden dimension of 512 and 4 attention heads per layer. The resulting denoiser has 62 million learnable parameters. Time and spatial information on the patches are added as input and we use a RoPE positional encoding [80]. Like in [37], input, output, and noise levels are preconditioned to improve the training dynamics. For sampling, we generate small trajectories of length 7 with 100 diffusion steps with a Adams-Bashforth multi-step sampler [87, 89].

**Evaluation metrics.** We use several metrics that can be computed between a sample and an observation. In evaluating magnetogram predictions, per-pixel averages are not informative since they are dominated by quiet Sun pixels even in patches [86, 28]. We therefore use multiple other metrics (see Tab. 1). First, the Wasserstein distance assesses the fit between the predicted distribution of pixels and the observed one. Second, we compute the mean absolute error in the isotropic power spectrum, which provides information on the spatial frequency content of an image. This metric is less sensitive to noise in the data. Finally, we consider physics-based summary statistics that characterize spatial gradients of the magnetic field. All metrics are averaged on all fields, on several realizations of the model, at several prediction dates, and averaged over several different time horizons.

**Baselines.** We compare our model to 4 baselines. Two fix the denoising architecture and compare the multiscale inference scheme with: an autoregressive inference scheme (a default choice in the literature) and the hierarchy-2 inference scheme from [25] (which sparsely completes missing frames, then autoregressively samples the remainder by conditioning on both past and future frames). The other two compare our model to existing spatiotemporal models for physical systems: [39] is a diffusion model tested on fluid dynamics data; and [50] is a deterministic transformer based on axial attention [30]. All models are trained with 40 epochs. We refer the reader to the Appendix for additional details.

**Solar predictions.** Tab. 1 confirms that, in this more challenging case, our multiscale inference scheme better predicts the pixel distributions than an autoregressive scheme at all future horizons (1:4, 4:16 and 16:32) by achieving the lowest Wasserstein distance. The spatial content is better preserved, shown by the error in the power spectrum, and illustrated in the predictions in Fig. 4. Our multiscale inference scheme also outperforms the "Hierarchy-2" model introduced for natural videos [25], which was not designed for slow-decaying, autocorrelated long-memory processes. Tab. 1

also shows that our diffusion model, equipped with our multiscale inference scheme, significantly outperforms existing models [50, 39]. A deterministic baseline such as AViT [50] can predict a future trajectory that is close to observed data but loses high frequency content, which gives rise to errors that accumulate with the rollout. Our model also compares favorably to the diffusion model of [39], which was developed for fluid dynamics data. These results showcase the limits of current models in probabilistic prediction of partially observable dynamical systems.

## 6 Conclusion and discussion

This work introduces and analyzes a multiscale inference scheme for predicting partially observable dynamical systems. Our approach efficiently incorporates past information—while being refined around the present—to predict future time steps. We show superior performance in both synthetic settings and the challenging task of predicting solar dynamics, outperforming existing schemes [25] and models [50, 39] for video and spatiotemporal physical systems prediction. Our results suggest that multiscale temporal conditioning helps mitigate partial observability, especially when long-range precursors influence future evolution, as in solar dynamics. To support further work, we contribute a dataset of high-resolution multi-modal solar regions trajectories.

While our method is well suited for long-memory systems with smoothly decaying temporal dependencies, it may not remain competitive when observations are dominated by short-term patterns. Future work could explore adaptive or learned conditioning strategies.

### Acknowledgments and Disclosure of Funding

The authors thank the Scientific Computing Core at the Flatiron Institute, a division of the Simons Foundation, for providing computational resources and support. They also thank Mark Cheung, Patrick Gallinari, Florentin Guth, and Ruoyu Wang for insightful discussions.

Polymathic AI acknowledges funding from the Simons Foundation and Schmidt Sciences.

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
