# OpenReview forum: "Predicting partially observable dynamical systems via diffusion models with a multiscale inference scheme"
_NeurIPS.cc/2025/Conference — NeurIPS 2025 poster_

### Official Review · Reviewer_EKRN · 2025-06-29

**Clarity:** 3
**Significance:** 1
**Originality:** 2
**Rating:** 3
**Confidence:** 3

**Summary:**

This paper presents an “inference scheme” for using diffusion models to model dynamical systems and applies it to a dataset from solar physics. The key challenge being addressed is that a diffusion model only predicts a fixed number of frames into the future, which could be much less than the desired prediction horizon. Naively, one could solve this problem by applying the diffusion model autoregressively. The authors proposed an alternative scheme where non-contiguous frames from the past are used to predict non-contiguous frames into the future. They argue that this incorporates long-term information better without increasing the compute cost. The demonstrate the advantage of this scheme on the solar dataset, which they will provide to the community.

**Questions:**

1. Can you be more explicit about why the scheme is suitable for stochastic, partially observable systems? One would imagine that the probabilistic nature of diffusion models can handle the former. For the latter, it’s also not clear why having access to more distant pasts necessarily helps. Suppose you have a dynamical system with a fast component coupled to a slow component and you only observe the slow component. It’s not clear why having longer-range interactions will help you deal with the unobserved fast component.

2. What is the design principle for the scheme? It may help to present a general algorithm for designing the scheme for various desired timescales and window sizes.

**Ethical Concerns:**

["NO or VERY MINOR ethics concerns only"]

**Final Justification:**

This work presents a novel scheme for using diffusion models to model temporal dynamics of physical systems. After back and forth with the authors, the assumptions about problems suitable for this approach are well stated (using autocorrelation instead of "long memory" is appreciated). The performance gain relative to the vanilla inference scheme is well supported by demonstrations on a solar physics dataset and a NS equation dataset. Nevertheless I am leaving the score at 3 (slightly below acceptance) to reflect the fact that I think for a NeurIPS paper one expects a broader scope.

**Limitations:**

The apparent specificity of the scheme for the analyzed dataset is a significant limitation. The authors should perhaps consider how the scheme is to be adapted for data with a wide range of timescales.

**Quality:**

3

**Strengths And Weaknesses:**

The writing is generally clear, and the proposed scheme appears to achieve substantial performance gains.
The weaknesses are mainly two fold. First, the motivations are not unclear. The authors stated (line 164) that some physical systems are challenging to model because they are stochastic and partially observable. These are well-known problems in dynamical system modeling in general and have well-honed solutions, such as probabilistic models with latent variables (e.g. hidden Markov model). The authors claim that they are addressing both issues, but it’s not clear how. Simply incorporating information from the more distant past does not solve these problems unless one makes additional assumptions (for example, that the latent variables have very slow dynamics). However, making these assumptions essentially collapses the three challenges (line 164-179) into the third one, “long-memory”, alone. In other words, it’s not clear why the proposed scheme is particularly suited to stochastic or partially observable systems.

The applicability of the scheme also appears limited. As the authors noted, the temporal templates need to be “hand-designed” and seem to strongly depend the nature of correlations in the data. This reviewer is not familiar with solar physics and thus cannot verify what is the typical timescale in the dataset, but in general for significance it is desirable if the proposed method is more general and can accommodate a wide range of timescales.

---

> ### Author Rebuttal · Authors · 2025-07-31
>
> We thank the reviewer for their feedback, especially regarding the motivation behind our approach and the types of physical systems we address, which has greatly helped improve the paper. Below, we respond to the main points concerning the motivation and applicability of our method.
>
> ----
>
> ## Question 1. Motivation of our multiscale templates
>
> Thanks to the reviewer’s point, we would like to clarify the following. The paper tackles probabilistic forecasting of a dynamical system, which has two main properties.
>
> - Partial observability. The state of the system at any given time cannot be fully determined by the observations. Consequently, the distribution of future scenarios conditioned on past observed data may not be restricted to a Dirac measure. It is this point that justifies the task of probabilistic forecasting, rather than deterministic forecasting.
>
> - Long memory. The system has slowly varying components that are important to model (e.g., the motion of plasma across several millions of kilometers on the Sun).
>
> Long memory alone is not sufficient to justify our method. For example, consider the synthetic case of Fig. 2 in our paper. It exhibits long memory, since it has a slowly varying sinusoidal component. If the process had zero stochasticity (making it fully observable), then all trajectories would exactly follow this smooth component, and a single state would be sufficient to determine the position on the sinusoid and what comes next. In that case, there would be no need for our multiscale templates.
> Now, if the system exhibits stochasticity, imagine we are at the negative peak (around $t=30$). Depending on the noise realization, the local trend may be positive or negative, making the state difficult to recover locally. In other words, the system’s partial observability, due to noise in the data, prevents accurate estimation of the slowly varying component. In this case, it is thus necessary to look further into the past, which is precisely what our multiscale inference scheme achieves, at no additional computational cost.
>
> This is supported by the experimental results reported below: as the noise level increases, the performance gain (measured as the ratio of Wasserstein distances; see Fig. 2) of our multiscale template becomes more pronounced. A ratio greater than 1 indicates improved performance with our model.
>
> | Noise level | 0.1 | 0.2 | 0.5 |
> |---|---|---|---|
> | Performance gain ratio | 2.71 | 3.95 | 4.28 |
>
> Following your comments, we would like to clarify that the “stochasticity” mentioned in the paper (line 164), such as measurement noise, can be considered part of the partial observability of the system. This has now been reflected in the abstract (line 9), which reads: “In this paper, we tackle the probabilistic prediction of partially observable dynamical systems with long-memory.”
>
> ----
>
> ## Question 2. Design principle
>
> The method is not specific to the Sun and accommodates a wide range of timescales, as shown in Fig. 4 (Appendix). The general algorithm is described in lines 227–242 in the main text. To improve clarity and highlight the generality of our approach, we will include an algorithm box in the paper.
>
> The multiscale templates are derived from first principles and are not hand-designed. The only dataset-specific adaptation lies in two hyperparameters: the time horizon that is generated at once (see Fig. 4, Appendix) and the number of future steps predicted per diffusion sampling process ($K$ in the paper). The later is driven by the computational budget allocated to the model (the more steps to generate at once, the larger the denoiser). The former depends on the time range of the dependencies. For example, if the system exhibits a maximum timescale $T$, after which any two observations $x_{t},x_{t+T}$ are independent, then this is a natural candidate for the horizon parameter. Otherwise, this can be tuned, and also provides a way to reveal the horizon of long-range dependencies in a dynamical system.
>
> To showcase the genereticity of our multiscale inference scheme, we applied it to the Navier-Stokes data from PDEArena [Gupta and Brandstetter 2022].
> To make the dynamical system partially observable, we downsampled it to 32x32 (from 128x128), considered only the density field (discarding velocity) and added a small Gaussian noise on the observations.
> The results on this new example of partially observable process are presented in the Table below with the same metric than in the paper (MAE on the power-spectrum, see line 296).
>
> | Inference scheme | 1–4 (MAE Power Spectrum) | 4–16 (MAE Power Spectrum) | 16–36 (MAE Power Spectrum) |
> |---|---|---|---|
> | Autoregressive | 0.39 | 0.43 | 0.35 |
> | Multiscale (ours) | **0.38** | **0.36** | **0.31** |
>
> We observe that our multiscale inference scheme improves predictions over the autoregressive baseline, particularly in the long-term interval (16–36), where it achieves a significantly lower MAE on the power spectrum. This shows that our approach is not specific to solar prediction. Rather, we considered solar forecasting as a challenging scientific application where our model also performs well.

---

> > ### Comment · Reviewer_EKRN · 2025-08-05
> >
> > My sincere thanks to the authors for responding to the rebuttal and addressing some of the concerns. The clarification about the design principle and the addition of another data example are both appreciated.
> >
> > However I still have concerns about the issue raised in Question 1. It is less about the results per se and more about how they are presented and framed with respect to the literature. The problem of modeling/predicting stochastic, partially observable and long-timescale systems is incredibly broad, both with in the AI for physics space and statistics in general. My concern is that the proposed scheme should not be presented as a general way of addressing this problem. Take stochasticity for example. To use the authors’ synthetic example in Sec. 5.1, suppose we have the system has noise that is i.i.d. over time (autocorrelation function is a Dirac’s delta around 0). In this case, it will not confer any benefit to have observations spaced far apart (as opposed to using adjacent frames). Spacing out the observations, as the proposed template does, is only beneficial against noise if we assume it to have autocorrelation that is positive and relatively long (relative to the timestep between frames). Similarly, there is an unobserved component that has very short timescales, then again I don’t think using spaced out frames will be better than contiguous ones. Is my intuition about these “thought experiments” correct?
> >
> > I'm happy to raise the score to a 3 at the current stage.

---

> > > ### Author Response · Authors · 2025-08-07
> > >
> > > We thank the reviewer for their time and valuable insights, which helped us refine the scope of our paper. In response, we have clarified two key aspects to better characterize the settings where our approach is typically most competitive, and we will revise the paper accordingly (see below).
> > >
> > > ---
> > >
> > > ## About stochasticity in the data
> > >
> > > Our synthetic example can be written as: $x_t = \lambda\mu_t + \epsilon_t$, where $\mu_t$ is a deterministic slow component, and $\epsilon_t$ is a random noise.
> > >
> > > We agree with you that, in the case of a Gaussian noise with no correlation, and in the absence of a slow component ($\lambda=0$), our method does not retain any advantage, for the reason you mention.
> > >
> > > However, when $\lambda>0$, the noise (even if decorrelated) makes it harder to "observe" the slow component $\mu_t$ locally.
> > > For example, at a low point of the slow sinusoidal trend (around $t=30$ in Fig. 2), some blue trajectories are locally increasing while others are locally decreasing, even though they are at the same point in the global trend.
> > > Hence, a model conditioned on a few contiguous time frames, misestimates the global trend and produces inaccurate future predictions.
> > >
> > > However, in a multiscale scheme, by looking further back in time, the model can "zoom out" from the high-frequency noise to better identify the slower component.
> > >
> > > Both your point and ours are actually confirmed by the experiment below on our synthetic example (now with decorrelated noise). We compute the same Wasserstein distance (see Fig. 2 caption) varying the amplitude $\lambda$ of the slow component from pure noise ($\lambda=0$) to a case where it is as dominant as in Fig. 2 ($\lambda=1$).
> > >
> > > | Amplitude $\lambda$ | 0.0 | 0.1 | 0.2 | 1.0 |
> > > |---|---|---|---|---|
> > > | Autoregressive    | **0.055** | 0.13 | 0.19 | 0.26 |
> > > | Multiscale (ours) | 0.060 | **0.027** | **0.036** | **0.067** |
> > >
> > > As we can see, when the decorrelated noise dominates ($\lambda=0$), a standard autoregressive contiguous scheme performs better. But when the noise masks a slow component ($\lambda\geq0.1$), our multiscale scheme is more effective at recovering it.
> > >
> > > ---
> > >
> > > ## Refining the notion of “Long-Memory”
> > >
> > > We agree that in the presence of a component at a very short timescale (typically the frame rate, or even shorter), the best approach is to condition on contiguous time frames.
> > > More generally, when the observations exhibit a dominant specific frequency (not necessarily the frame rate), we do not expect our approach to outperform conditioning on uniformly sampled frames at the corresponding frequency.
> > >
> > > However, in many real-world settings, such as solar and fluid dynamics, observations exhibit a smooth decay of their autocorrelation (sometimes characterized in the frequency domain by a power-law decay of the power spectrum; see [53], Chapter 9 on long memory, or [2, 49, 56]), a setting for which our multiscale inference scheme is specifically suited.
> > > This excludes the cases mentioned above, involving short-timescale patterns or strong periodicity, which would produce spikes in the autocorrelation. In the case of the Sun, we do observe such a smooth decay on the timescales of interest (see Fig. 3 in the Appendix).
> > >
> > > This notion is more precise than the term "long-memory" used in our paper and can be tested empirically.
> > > To better convey the scope of our paper, we will revise it as follows:
> > > - l8. (Abstract). "In this paper, we address the probabilistic prediction of partially observable dynamical systems with long memory, characterized by a smooth decay of their autocorrelation, with applications to solar dynamics and the evolution of active regions."
> > > - l175. (Method section). "Many physical systems exhibit long memory, or long-range dependency, in time. This can be characterized by a smooth decay of the autocorrelation (sometimes characterized in the frequency domain by a power-law decay of the power spectrum [53, 2, 49, 56]). Intuitively, states closer to the present have a stronger impact on the future and the influence of increasingly distant past states gradually diminishes while remaining significant."
> > > - l330. (Discussion) "While our method is well suited for long-memory systems with smoothly decaying temporal dependencies, it may struggle to remain competitive when observations are dominated by short-term patterns, for example. Future work could explore adaptive or learned conditioning strategies."

---

### Official Review · Reviewer_xJBu · 2025-07-02

**Clarity:** 3
**Significance:** 3
**Originality:** 2
**Rating:** 5
**Confidence:** 3

**Summary:**

The paper is concerned with the problem of estimating the state of a dynamical system when the measurements only contain a subset of the states, referred to as partially observable dynamical systems. The motivation comes from physics and more specifically a problem of solar prediction. The main development of the paper is an inference scheme that is capable of working on several scales and in that way introduce a mechanism for including long-range dependencies. The approach is tested on a large dataset of videos on solar activity. The data used for the experiments will be  made available to the community to ensure reproducibility of the research.

**Questions:**

1. In many real-world situations the measurements are available as functions of the state. Would you be able to extent to that case or is that something that requires significant effort?
2. In an effort to reduce the connection to the solar application and shift the focus more to the generality of the development, would it be possible to compare your developments on the test cases provided here
https://arxiv.org/abs/2011.04006
referred to as the long range arena? The reason for this question is that these problems have become quite popular when it comes to profiling algorithms focusing on long range dependencies.
3. More of a comment really, but by using a state-space construction you should be able to access the smoothing distribution (5) without risking the computational complexity to blow up.

**Ethical Concerns:**

["NO or VERY MINOR ethics concerns only"]

**Final Justification:**

The authors responded very well to my questions and resolved the concerns I communicated in my questions in a good and convincing manner. Very nice that they also offered a new example on such short notice.

**Limitations:**

Yes

**Paper Formatting Concerns:**

None.

**Quality:**

2

**Strengths And Weaknesses:**

Strengths
* The paper is well-written and easy to read.
* The development is intuitive and comes across as natural.
* Very good to see that the authors are making the data available. This is necessary for the results to be reproducible and it can also be a resource for other researchers, in particular those with an interest in this particular application.

Weaknesses
* Miss relationship to the state space model and in particular the so-called structured state space model, see e.g.
https://arxiv.org/abs/2111.00396
* Would have been good to see your performance on some of the existing long-range tasks that are available in order to gauge the performance compared to existing approaches. Note that this does not require you to implement an existing approaches, it is enough to run your method on these tasks.
* Perhaps too much focus on the solar application.

---

> ### Author Rebuttal · Authors · 2025-07-31
>
> We thank the reviewer for their insightful comments and suggestions, and for highlighting both the clarity of the manuscript and the value of releasing the solar dataset to the community.
>
> Below are our answers to your questions.
>
> ---
>
> ## Question 1
>
> Yes, we can extend to the case where the observations are available as functions of the state without a lot of effort, so long as the function in question is differentiable. Indeed, in that case, one can learn a diffusion model on the states themselves through an expectation-maximization algorithm for example [Rozet et al., 2024]. Then, one can apply our multiscale inference to the diffusion model in the same way as in our paper. The expectation-maximization algorithm and the inference scheme are essentially orthogonal.
>
> ---
>
> ## Question 2
>
> While interesting, the tasks introduced in the Long Range Arena benchmark [Tay, Dehghani et al., 2020] focus on compositionality, document similarity, and classification, and thus fall outside the scope of our work, which addresses probabilistic forecasting of partially observable dynamical systems.
> Since we agree that validating our method beyond the solar problem is important, we applied it to the PDEArena dataset [Gupta et al., 2022], specifically to the Navier-Stokes trajectories (see their Fig. 1). To make this system partially observable, we downsampled it from 128×128 to 32×32, retained only the density field (discarding velocity), and added small Gaussian noise to the observations.
> Results for this new partially observable system are reported in the table below, using the same metric as in our paper (MAE on the power spectrum, see line 296).
>
> | Inference scheme | 1–4 (MAE Power Spectrum) | 4–16 (MAE Power Spectrum) | 16–36 (MAE Power Spectrum) |
> |---|---|---|---|
> | Autoregressive | 0.39 | 0.43 | 0.35 |
> | Multiscale (ours) | **0.38** | **0.36** | **0.31** |
>
> We observe that our multiscale inference scheme improves predictions over the autoregressive baseline, particularly in the long-term interval (16–36), where it achieves a significantly lower MAE on the power spectrum. This shows that our approach is not specific to solar prediction. Rather, we considered solar forecasting as a challenging scientific application where our model also performs well.
>
> ---
>
> ## Question 3
>
> Thank you for the insightful comment. We now include a discussion of state-space models in the "Related Works" section. We would like to emphasize that the focus of our paper is not on the choice of architecture, but rather on how to perform prediction within a conditional diffusion model. Diffusion models are increasingly being applied to dynamical systems [63, 67, 73, 34], and have demonstrated strong performance in generative modeling, which aligns well with our task: probabilistic forecasting.
>
> ---
>
> Our paper does focus on the solar prediction problem (although we also include a synthetic example and will add the results on PDEArena), because it is currently a very challenging task in probabilistic prediction, with the potential to significantly advance solar flare prediction in the future [41, 10, 43].

---

> > ### Comment · Reviewer_xJBu · 2025-08-05
> > **Response to the rebuttal**
> >
> > Thank you for your response. It is clear and to the point and resolves the questions I raised in a good way. I very much appreciate your response to my second question. Really nice and impressive that I managed to provide this example on such short notice. I have reread your paper in light of the response you gave to my questions and I am raising my overall score from 3 to 5.

---

> > > ### Author Response · Authors · 2025-08-07
> > >
> > > Thank you for your follow-up and for raising your score. We are pleased that our response addressed your questions. Please don’t hesitate to reach out if you have any further inquiries. Thank you again for your valuable suggestions.

---

### Official Review · Reviewer_Ac1G · 2025-07-02

**Clarity:** 2
**Significance:** 3
**Originality:** 3
**Rating:** 5
**Confidence:** 3

**Summary:**

In this paper, authors propose a multiscale inference scheme for diffusion models to solve the problem of insufficient information for prediction. The model can capture long-range temporal dependencies by generating trajectories that are temporally fine-grained near the given timestamp while coarser when moving farther away. The effectiveness of the proposed model has been evaluated on the solar prediction task, and authors also build a corresponding large multi-modal dataset.

**Questions:**

Please refer to the Weaknesses part.

**Ethical Concerns:**

["NO or VERY MINOR ethics concerns only"]

**Final Justification:**

The multiscale temporal templates proposed in this paper are interesting and reasonable. Although initially confusing due to some unclear descriptions and typos, the authors' explanations have made things much clearer, and with the appropriate revisions, it should be easier to understand the underlying process. This multiscale template takes into account both short-term and long-term dynamics, and uses the generated long-term future step as a condition to complete the intermediate steps, thus ensuring temporal continuity. The overall design is sound and supported by experiments. Taking all factors into consideration, I think this paper exceeds the acceptance bar of NeurIPS, and I have increased my rating from 4 to 5.

**Limitations:**

Yes.

**Paper Formatting Concerns:**

No.

**Quality:**

3

**Strengths And Weaknesses:**

Strengths:
1. Authors incorporate multiscale temporal templates to allow the model to focus on both short-term and long-term dynamics without increasing computational cost, breaking the limitations of traditional autoregressive rolling prediction that gradually ignores the ground-true historical observation and accumulates errors.
2. The design of multiscale templates is flexible and controllable, which provides the possibility of easy deployment for real applications.
3. The effectiveness of the proposed method has been sufficiently evaluated on both synthetic and real datasets, along with ablation studies and analyses. Besides, authors prepare a large-scale solar dataset, which should be helpful for the corresponding community.

Weaknesses:
1. The temporal template chosen on Sec.4.2 is a little confusing. In the first round, {-9, -3, -1, 0, 1, 3, 9} is chosen and history (<=0) is assumed to be fully available, i.e., history from t=-9 to t=0 should be available, then in the second round, why not directly choosing {-5, -4, -2, 0, 2, 4, 5} directly but using shift? Besides, after shifting the given {-6, -2, -1, 0, 1, 2, 6} 2 steps, it should be {-4, 0, 1, 2, 4, 8}, the overlap with previously generated steps is not {-3, 1, 3}.
2. Since the temporal template is not evenly sampled, can DiT well capture such flexible temporal relationships?
3. As authors claimed, the paper aims at solving the solar prediction, besides diffusion-based methods, other temporal models designed for the same task should be involved as baselines.

---

> ### Author Rebuttal · Authors · 2025-07-31
>
> We thank the reviewer for their positive feedback on our work, particularly for recognizing that our method is "breaking the limitations of traditional autoregressive rolling prediction" and for highlighting its "possibility of easy deployment for real applications".
>
> ---
>
> ## Question 1. Part 1.
>
> We are not directly choosing the template {-5, -4, -2, 0, 2, 4, 5} because it is not, strictly speaking, a multiscale template in the sense of Definition Eq. 7 in our paper. This template is refined near $t = -5$ and $t = 5$, but not around $t = 0$, whereas the motivation behind our multiscale templates is to provide finer resolution around $t = 0$ and become progressively coarser farther from it (see line 199).
> Moreover, if we were to first use the template {-9, -3, -1, 0, 1, 3, 9} and then switch to {-5, -4, -2, 0, 2, 4, 5} as suggested, this would mean that future steps {1, 3, 9} are generated before steps {2, 4, 5}, potentially introducing temporal discontinuities. For instance, the model could generate a scenario where {1, 3, 9} become more active, while {2, 4, 5} become less active — even though both sets are conditioned on the same past steps {-5, -4, -2, 0}. This is precisely why our multiscale scheme (see Fig. 1, right) is designed to always condition on already generated future steps, thereby preserving temporal coherence.
>
> ---
>
> ## Question 1. Part 2.
>
> Indeed, the second multiscale template {-6,-2,-1,0,1,2,6} should be shifted by 3 (not by 2) to obtain {-3,1,2,3,4,5,9}, which overlaps with the steps {-3,1,3}. These correspond to either past steps {-3} or already generated future steps {1,3}. This has been updated in the manuscript (lines 234–235).
>
> ---
>
> ## Question 2
>
> Handling unevenly sampled trajectories is primarily achieved by concatenating time information to the input of our denoiser, as described on line 289 of the paper. More precisely, we add an additional channel to the input tensor that indicates, for each pixel in the input trajectory, its relative time index (e.g., -9, -3, -1, 0, 1, 3, 9). Further details are provided in Appendix (lines 99–104)
>
> ---
>
> ## Question 3
>
> Table 1 in our paper reports the results of a non-diffusion-based architecture, specifically a recent transformer architecture designed for next token prediction [52], that has been consistently used for the prediction of physical dynamical systems.
> Below are the results for such baseline, as reported in our paper. Note that published models on solar prediction [1,66] operate in substantially worse spatiotemporal resolutions (e.g., a time resolution of 12h while we predict hourly, or 2x spatially downsampled compared to our data) than this baseline.
>
> | Model | Model type | Wasserstein 1:4 | Wasserstein 4:16 | Wasserstein 16:32 | PS MAE 1:4 | PS MAE 4:16 | PS MAE 16:32 |
> |---|---|---|---|---|---|---|---|
> | AViT [52]    | Transformer | 12 | 13 | 15 | **0.11** | 0.35 | 0.81 |
> | DiT (ours)   | Diffusion   | **3.0** | **4.3** | **5.5** | 0.12 | **0.22** | **0.33** |
>
> Note that a transformer, as any deterministic baseline, will suffer from the well-known smoothing effect, as shown below. The transformer model produces a prediction with significantly less power in the high-frequencies than the observed spectral content (measured as the sum of the power-spectrum (PS) over the top 75% high frequencies).
>
> | Model | Model type | PS relative energy (low frequencies) | PS relative energy (high frequencies) |
> |---|---|---|---|
> | AViT [52]    | Transformer | -0.89\% | -82\% |
> | DiT (ours)   | Diffusion   |  -3.8\% | +17\% |
>
> This smoothing behavior would occur with any deterministic baseline, and we preferred to focus our paper on comparing different schemes within the same diffusion model.

---

> ### Comment · Reviewer_Ac1G · 2025-08-02
>
> Thank you for your response. Your explanation has addressed most of my concerns, and I now have a better understanding of your motivation and the rationale behind your template choice. After careful consideration, I think this paper is above the acceptance bar of NeurIPS, and I'm willing to increase my rating from 4 to 5. Good luck.

---

> > ### Author Response · Authors · 2025-08-04
> >
> > Thank you for the follow-up and for raising your score. We're glad our response addressed your concerns.
> > If you have any additional questions, feel free to let us know!

---

### Official Review · Reviewer_1zXs · 2025-07-02

**Clarity:** 2
**Significance:** 2
**Originality:** 2
**Rating:** 5
**Confidence:** 2

**Summary:**

The paper proposes a method to train a conditional diffusion model to predict future states in a dynamical system, while being able to learn and exploit long-range dependencies in the data remaining computationally efficient. The method is motivated by a problem in solar physics, for which an associated dataset will be released also.

**Questions:**

I suggest the authors to write down in what sense their generated samples can be seen as approximating the distribution of interest, and how making certain Markovian of order k assumptions about the data would relate to the adaptive choice of $\alpha$ in the methodology.
Finally, I express a good level uncertainty as I am not familiar with the baselines involved here.

**Ethical Concerns:**

["NO or VERY MINOR ethics concerns only"]

**Final Justification:**

See below comment for more details. I cautiously increase my score due to the potential improvements that can be made to the paper from the rebutal(s), yet I remain with low confidence due to my lack of experience with these type of approaches.

**Limitations:**

The authors do discuss limitations.

**Quality:**

2

**Strengths And Weaknesses:**

Strenghts: the paper has a strong application; it does not involve "mathiness", which is to be appreciated these days; presents interesting and relatively well explained results.

Weaknesses:

**[Connections with related prediction approaches in dynamical systems are not very clear.]** Firstly, the field with which I am most familiar with, data assimilation (DA), is cited/mentioned in lines 182-183, but the connection remains unclear. The authors also cite DA works in lines 79-81, stating: " Diffusion models can also be used for data assimilation and prediction from incomplete observations [67, 73, 34], but this is an orthogonal direction of work, and a dataset of fully observed states from the system must exist to train the model." It is not clear to me whether the last part of the sentence refers to the authors' work or to the cited DA work. In DA or Bayesian filtering, the "states" are not observed - they are latent - typically called x_t and only y_t is observed; partial observability in DA/filtering is the whole point. I guess it is not clear what "partial observability" means for the authors here. In DA, it is very mathematically defined by the framework of probabilistic state space models (see e.g., Kevin P. Murphy's book on probabilistic machine learning, section on state-space models, or Sarkka & Svensson 2023 "Bayesian filtering and smoothing").

**[Clarifications needed regarding assumption (b) in Sec 4.1]** : continuing really from the point above; authors claim that " Diffusion models have generally been applied to dynamical systems without fully addressing challenge (b) " but cite some DA works including eg [67] that do exactly this, in the meaning of partial observability. This point also adds confusion by mixing PO with Markovianity. Although the authors never state this fully explicitly (or clearly), they must be assuming non-first order Markovianity of their observations here, otherwise it would clearly be pointless to condition on anything beyond the previous state. The authors at some point mention "noise in the data" in the experiments, although this seems not explicitly modelled, which again leaves me confused.

Overall, Eq (6) leaves a "handwavy" feel with respect to the statistical modelling done here, and it is not clearly explicited in mathematics what is assumed about the underlying data generating process. Again it is ok to use diffusion models and neural nets and this has been done in related works in data assimilation, but it has to be clearly explicited which parts of the dynamical system is the score modelling/related to, for example.  Section 3 feels isolated and disconnected from the rest, mathematics wise.

---

> ### Author Rebuttal · Authors · 2025-07-31
>
> We thank the reviewer for their positive feedback, in particular for acknowledging the strength of the application, the readability of our paper, and for the valuable insights on data assimilation (DA).
>
> ----
>
> ##  Connections with related prediction approaches in dynamical systems
>
> Although we follow developments in data assimilation (DA) with interest, our paper does not directly address this topic. Instead, it focuses on probabilistic prediction of the observations themselves, without attempting to reconstruct the system's underlying states. We updated the “Related Works” section based on your comment about data assimilation (DA).
> - Line 68 now reads: “our work falls within the scope of modeling the dynamics of the observations”.
> - Line 76 now reads: “Thus, while [40, 63] see no benefit from using more than two past observations, incorporating additional past steps substantially improves results in our setting”.
> - Line 79 now reads: “Diffusion models can perform data assimilation and prediction from incomplete observations simultaneously [67, 73, 34], but this requires a dataset of underlying system states to train the model -- an assumption we do not make in this paper”, where the last part of the sentence refers to [67, 73, 34].
>
> In our paper, partial observability is defined by the fact that, at any time, the conditional distribution of the state given its observation is not restricted to a Dirac measure, characterizing the impossibility of reconstructing the system’s state from the observation. This will be stated more clearly in the introduction and in section 4.1. In that sense, we align with the standard mathematical framework used in Bayesian filtering and DA, without explicitly writing out the equations, since, as you mention, we aimed to avoid any “mathiness” not directly useful for developing our approach. In the solar prediction application tackled in our paper, there are substantial unknown latent physical variables (e.g., all state variables of the interior, the magnetic field in the Sun's atmosphere) as well as open questions about dynamics (e.g., how the atmosphere of the Sun is actually heated) and entirely missing subgrid physics that cannot be observed in our data.
>
> ----
>
> ## Clarifications needed regarding assumption (b) in Sec 4.1
>
> We thank the reviewer for pointing out this sentence. It has been replaced by: \
> "
> Diffusion models have been applied to prediction on dynamical systems by using additional methods to remove assumption (b). For example, [40] apply a diffusion model to fully resolved fluids which are effectively Markovian. In weather prediction, although the observed data are sparse, data assimilation -- also known as reanalysis -- enables the reconstruction of missing information, resulting in large datasets of highly informative states [28], on which diffusion models have been successfully trained [63]. Other models handle missing states, but require clean sates for training [73,34], which is not always available.
>
> In this paper, we tackle the challenging problem of predicting a dynamical system that presents the three difficulties (a), (b), and (c) simultaneously, without assuming that (b) can be circumvented, as is often the case in many disciplines.
> "
>
> ----
>
> ## Other clarifications
>
> As you point out, the system has to be non-first-order Markov for our method to show an advantage over autoregressive systems. We did not mention this assumption as we allow ourselves to test it empirically. In practice, if a standard autoregressive scheme outperforms our multiscale approach, this may suggest that the process is Markovian of some order.
>
> Indeed, it is important to clarify that we learn the score of each of the distribution $p(x_{I_n}| x_{C_n})$ in Eq.6 in the same way as [Harvey et al., 2022] did, by training a denoiser which takes as input the relative positions of the time steps in the context $C_n \cup I_n$ (see line 289-290 and Appendix, line 99-104).
>
> Section 3 will be renamed "Background: Conditional Diffusion Models".
>
> ---
>
> ## Questions
>
> As stated above, we learn the score of the distributions $p(x_{I_n}| x_{C_n})$, which can be used to approximate the distribution $p(x_{1:T} | x_{t\leq0})$ (see Eq. 6). Let us recall that we assume the set of indices $I_n$ and $C_n$ to have $K$ and $K+1$ elements respectively.
>
> In certain restricted cases, one can prove that Eq. 6 is actually an equality, for example, under a Markovian assumption of order $K+1$, and employing a standard autoregressive inference scheme (see line 155 in our paper). In that case, it makes sense to model the target distribution $p(x_{1:T} | x_{t\leq0})$ by learning the score of $p(x_{I_n}| x_{C_n})$ for all $n$.
>
> Studying this approximation in the case of generic $I_n,C_n$ is hard, and it is not clear that a Markovianity assumption of order $K+1$ is well suited for justifying the choice of the maximum $\alpha$ in our inference scheme. At this stage, a proper explanation would require a separate LaTeX note, which is not permitted this year.
>
> Intuitively, if we assume that the process has a maximum timescale of dependence, that is, there exists a time $T$ such that for any time $\tau\geq T$, the variables $x_t$ and $x_{t+\tau}$ are independent (often referred to as the integral scale in signal processing), then it is natural to restrict the extent of the largest multiscale template $\mathbb{T}^{\alpha_{max}}_K$ (defined in Eq. 7) to be smaller than $T$, since there is no need to model dependencies beyond $T$. However, this intuition would need to be made more rigorous.

---

### Note · Authors · 2025-08-12

We are grateful to the reviewers for their insightful questions, which helped us clarify our method, scope, and relation to prior work. As a result, three reviewers raised their scores to 5 (accept).

In our latest message to Reviewer EKRN, we responded to their remaining point concerning how the “results [...] are presented and framed with respect to the literature.” Specifically, we proposed clarifying the use of the term “Long-Memory” throughout the paper and expanding the discussion of limitations (see the suggested text changes in our response). We appreciate the reviewer’s thoughtful feedback, which helped improve our work.

We believe this final point has been adequately addressed in our response and does not impact the main contributions of our paper.

---

### Decision · Program_Chairs · 2025-09-17

**Decision:**

Accept (poster)

**Comment:**

The paper addresses the problem of prediction in partially observable dynamical systems. It proposes an inference scheme based on diffusion models that is able to predict future frames in a multiscale fashion by conditioning also on multiscale past frames. The idea here is that, compared to naive autoregressive prediction approach, multiscale inference can better capture long-range dependencies without sacrificing computation cost. The approach is tested on a large dataset of videos on solar activity. This dataset is very interesting in its own right, and the authors promise that they will make it available to the community.

This is a well written paper that deals with an interesting and challenging application in solar prediction. It develops a sensible multiscale diffusion inference method. The reviewers gave many useful comments that the authors should address when updating their paper.